# Retention of Improved Plantar Sensation in Patients with Type II Diabetes Mellitus and Sensory Peripheral Neuropathy after One Month of Vibrating Insole Therapy: A Pilot Study

**DOI:** 10.3390/s24103131

**Published:** 2024-05-15

**Authors:** Liezel Ennion, Juha M. Hijmans

**Affiliations:** 1Department of Physiotherapy, University of the Western Cape (UWC), 10 Blanckenberg Road, Bellville, Cape Town 7530, South Africa; 2Department of Rehabilitation Medicine, University Medical Center Groningen, University of Groningen, Hanzeplein 1, 9713 GZ Groningen, The Netherlands; j.m.hijmans@umcg.nl

**Keywords:** sensory peripheral neuropathy, therapeutic, vibration

## Abstract

Sensory peripheral neuropathy is a common complication of diabetes mellitus and the biggest risk factor for diabetic foot ulcers. There is currently no available treatment that can reverse sensory loss in the diabetic population. The application of mechanical noise has been shown to improve vibration perception threshold or plantar sensation (through stochastic resonance) in the short term, but the therapeutic use, and longer-term effects have not been explored. In this study, vibrating insoles were therapeutically used by 22 participants, for 30 min per day, on a daily basis, for a month by persons with diabetic sensory peripheral neuropathy. The therapeutic application of vibrating insoles in this cohort significantly improved VPT by an average of 8.5 V (*p* = 0.001) post-intervention and 8.2 V (*p* < 0.001) post-washout. This statistically and clinically relevant improvement can play a role in protection against diabetic foot ulcers and the delay of subsequent lower-extremity amputation.

## 1. Introduction

In 2021, 537 million adults globally were diagnosed with diabetes mellitus (DM) and 6.7 million people died as a result of the complications of DM [1]. Another 541 million people have Impaired Glucose Tolerance (IGT), placing them at high risk for developing type II DM in the near future [1]. Three out of every four people diagnosed with DM live in middle- and lower-income countries such as South Africa [1]. Of these, 66.8% of cases are currently undiagnosed [1], and approximately 60% of all patients diagnosed with DM will develop diabetic peripheral neuropathy (DPN) [2]. 

One of the most common symptoms of peripheral neuropathy is loss of cutaneous sensation in the feet [2]. Having sensory loss is the biggest risk factor for developing a diabetic foot ulcer. The lifetime risk for developing a foot ulcer is 15–20% in the diabetic population, and once patients with diabetes develop a foot ulcer, 34% will suffer a lower limb amputation within the subsequent year [1,3,4]. Currently, there is no medication or therapy that has been proven to improve or reverse sensory loss in persons with diabetic sensory peripheral neuropathy over the long term.

Vibration perception threshold (VPT) is a measure of the severity of neuropathy. The application of mechanical noise has proven to be effective in improving VPT in the short term through a mechanism called stochastic resonance [5]. In this study, conducted in the Netherlands, mechanical noise was mainly used to improve plantar sensation during the actual use of vibrating insoles [5]. In other, similar studies, the improvement in VPT remained after the prolonged use (up to 60 min) of vibrating insoles, suggesting that no adaptations of the mechanoreceptors took place [6,7]. The lack of adaptation identified in these studies (67) suggested the potential of applying mechanical noise repeatedly over a longer period of time. Long-term application might provide the CNS with more tactile input, which might result in longer-lasting effects. To date, the effect of repeated application over a longer period and the longer-term retention of the effects have not been explored, other than in a single case study [8]. The current study aimed to determine if the repetitive therapeutic application of mechanical noise applied to the plantar surface of persons with Type II DM could improve VPT and if the improvement was maintained over time in a bigger sample.

## 2. Methods

A prospective, longitudinal, quasi-experimental design was utilised to collect data for the study. The research was conducted in the Cape Town metropolitan area of Western Cape province, South Africa. Ethics approval was obtained from the University of the Western Cape’s Biomedical Research Ethics Committee (BM16/3/23). Permission to conduct the study was obtained from the Western Cape Provincial Department of Health and the respective facilities.

Participants were recruited from public community health centres (CHC) in Cape Town, two private diabetic clinics, and through snowball sampling, where participants referred other potential participants. The relevant medical history of the referred patients was obtained and potential participants were evaluated against the inclusion criteria.

Of the 90 participants screened, 31 met the inclusion criteria and were enrolled in the study. The inclusion criteria were as follows: age 18–80 years; type II diabetes and diabetic SPN; no neurological disease unrelated to DM, no diagnosis of tuberculosis or human immunodeficiency virus (HIV), and a mean vibration perception threshold (VPT) measurement of 20 V–50 V on at least one foot, measured with the Bio-thesiometer (Bio-thesiometer USA, Serial nr. 6694 LP) at seven points recommended in the Bio-thesiometer manual (tip of first toe, base of first toe, tip of second to fifth toes, and middle of the arch). Nine participants did not return for follow-up assessments and were excluded from the study. The final study sample consisted of 22 participants.

The Bio-thesiometer measurements at the seven previously mentioned locations were repeated three times. The mean score for each point was calculated and then averaged across the seven points. The reported VPT score for the participant was the mean value of both feet. Baseline measurements of VPT were taken by an independent research assistant prior to the intervention (VPT 1), following the intervention (VPT 2), and after a one-month washout period (VPT 3).

The intervention was provided by a vibrating insole (SureStep™, Phoenix Medical Technologies, Providence, RI, USA) and applied for 30 min daily by the participants for one calendar month. The participant was in a seated position with feet placed on the insoles. The insoles transmitted a non-invasive, mechanical vibration to the plantar aspect of the feet. The stimulus waveform consisted of a white noise bandpass filtered to 50–500 Hz [9]. The vibration was generated by three piezoelectric actuators located at the medial and lateral metatarsal phalangeal joints and heel. The amplitude of the vibration was determined by adjusting the volume level (output voltage) of an MP3 player connected to the insoles. The initial amplitude of the vibratory stimulus for each patient was set at a supra-threshold level. Participants were instructed to maintain the vibration amplitude at the supra-threshold and to keep a daily log to document compliance and changes in the amplitude of the MP3 player’s volume setting. The insoles were well-tolerated and no adverse effects were reported during the study. Figure 1 shows an overview of the intervention.

A data analysis was performed using SPSS (version 27). Descriptive statistics of the participants were recorded. To determine the effectiveness of the intervention, a repeated measures ANOVA was performed. When a main effect of time was found, paired *t*-tests with Bonferroni correction (alfa = 0.05/3) were performed to determine the differences between the three time points.

## 3. Results

The mean age of the participants was 58 years (range 39–78; SD ± 9 years and 11 months). Fourteen females and eight males participated in the study. A main effect of time on VPT was shown (F(1, 1.738) = F 248.004, *p* < 0.001)). The reduced VPT effect was present post-intervention (VPT2) in 86% (N = 19) and in 91% of participants (N = 20) post-washout (VPT3) (Table 1). Pairwise comparisons show significant changes between pre-intervention and post-intervention and between pre-intervention and post-washout. No difference was found between post-intervention and post-washout (Table 1 and Figure 2), indicating that the therapeutic effect was retained over time.

According to their self-reported treatment logs, all 22 participants who were included in the final sample were compliant with the daily use of the device for 30 min.

## 4. Discussion

There are no known non-pharmacological interventions that lead to a retention of the short-term efficacy in reducing VPT in people with diabetes [10]. Following a single case-study (with these insoles) [8], this is the first intervention shown to improve VPT in the diabetic population and retain this effect after a washout period of one month.

The mechanism of this improvement is poorly understood. One of the potential working mechanisms which could explain the immediate effects of mechanical noise stimulation of the plantar side of the foot is stochastic resonance. Through stochastic resonance, previously unfelt subthreshold stimuli become suprathreshold stimuli and produce increased action potentials which would not occur without the noise signal [5]. This could explain why subthreshold mechanical noise stimulation can improve standing balance [11], gait [12], and tactile sensation [13] in the short term. However, all these effects are immediate, short-term effects while the noise stimulation is being applied. In the current study, we showed long-lasting effects (improved VPT) in the absence of active noise stimulation. Potentially, neuroplasticity of the brain could play a role, where increased tactile information (due to more action potentials) during the therapy may cause reorganisation in the brain, improving the perception of tactile stimulation. However, other non-stochastic stimuli, like electrical stimulation, have also proven to be effective in improving tactile sensation [14]. Therefore, it is questionable whether stochastic resonance or neuronal reorganisation is the only working mechanism to explain the improved sensation.

Even though other working mechanisms may explain the long-term effects of mechanical noise stimulation, the use of random noise as a stimulus instead of a vibratory stimulus with a fixed frequency could have other positive effects on the results. It is known that adaptation takes place in mechanoreceptors when they are stimulated with a consistent-frequency vibration [15]. This adaptation would negatively impact sensation by increasing the threshold for tactile sensation and therefore decreasing the sensitivity. Seemingly, the mechanoreceptors do not adapt to the application of random noise, so the potential negative effects of a consistent vibratory stimulus on tactile sensation are eliminated, potentially resulting in improved sensation.

The current study showed that the daily (therapeutic) use of vibrating insoles for 30 consecutive days, 30 min/day, reduced the vibration perception threshold by 8.5 V in a cohort of diabetic patients with sensory peripheral neuropathy. The intervention was well tolerated with a self-reported compliance rate of 100%. In many cases, participants were reluctant to return the device because they enjoyed the sensation and noticed a marked improvement in their pedal sensation. Previous studies have shown that there is a strong relation between VPT and ulceration rates [16]. The Vibration Perception Threshold can predict those patients with diabetes who are at increased risk of foot ulceration and a VPT > 25 V carries a sevenfold higher risk of foot ulceration [16]. Further, a multicentre study found that in patients with a VPT > 25 V, for each 1 V increase in VPT, the risk of foot ulceration increased by 5.6% [17].

Significantly, an 8.5 V reduction in VPT was observed following the intervention, which persisted (V reduction) after a washout period of one month. Therefore, the 8.5 V reduction in VPT demonstrated in the current study implies that if the inverse relationship between VPT and risk of ulceration is also valid, the vibrating insoles could lower the risk of foot ulceration amongst high-risk diabetics by as much as 47%. This suggests that the vibrating insoles may be used therapeutically in patients with diabetic SPN, potentially resulting in a protective effect against foot ulceration.

If the onset of a first ulcer can be avoided or delayed, there would be considerable medical cost savings and the risk for subsequent lower-extremity amputation would be significantly lower [18,19]. It is commonly known that sensory loss is progressive in the diabetic population. Therefore, it is unlikely that the observed improvement in VPT could be attributable to causes other than the intervention.

The continuous improvement in VPT one month after the intervention was stopped was not explored in this study and cannot currently be explained clearly. However, the fact the VPT was still improved after one month of no intervention when compared to the baseline, and in some cases was even better than post-intervention results, could allude to a systemic change. This hypothesis could potentially be supported by the findings of studies utilising full-body vibration [20,21]. In both these studies, whole-body vibration resulted in improved tissue oxygenation and skin blood flow [20,21]. Similarly, this effect (improved skin blood flow) was also noted and retained for a period of time in a more recent study where 35 Hz vibration was applied locally to the plantar aspect of the feet of patients with diabetes mellitus [22]. Improved tissue oxygenation and skin blood flow could lead to improved peripheral nerve health and subsequently improve sensation, as observed in the cohort of the current study utilising vibrating insoles.

This theory could potentially explain the retention of the effect over time, but further study on this topic is warranted.

Another hypothesis to try and explain the retention of improved vibration perception threshold over time is that the effect could be attributed to the increased presence of nitric oxide (NO). It is commonly known that NO production is impaired in patients with diabetes [23]. Nitric oxide release is stimulated by insulin [24]. As insulin production is reduced or impaired in patients with diabetes mellitus, there is a marked reduction in the presence of NO in this population [24]. Nitric oxide plays an important role in regulating the vascular system [24].

Impaired NO production and its presence in the vascular system can result in hypertension, cardiovascular disease, angiogenesis-associated disorders and atherosclerosis [25,26,27]. Nitric oxide is also responsible for the secretion of vascular endothelial growth factor (VEGF), which, in turn, stimulates angiogenesis [28].

Therefore, inversely, the presence or increased production of NO can be considered protective against these vascular complications, commonly identified in patients with type II DM.

In music studies, low-frequency vibrations (similar to what the insoles produce) have also been proven to stimulate the release of NO [29]. The oscillating low-frequency vibrations produced by the insoles could possibly have led to the increased production of NO, which, in turn, would lead to vasodilation and improved skin blood flow and tissue oxygenation, as well as angiogenesis, all of which could improve nerve health and regeneration. The increase in NO production specifically upon lower-frequency vibrations might explain the longer-term improvement in vibration perception threshold in this cohort, but further investigation is necessary to prove this hypothesis.

There are some limitations to this study. No control group was used in this study. Future studies adopting a control group receiving a sham intervention should be performed to verify the results of this study. A second limitation was that compliance was not recorded objectively. Although the results are very promising, actual use of the insoles should be objectively recorded in future studies in order to see whether there is a relation between the actual therapy time and the level of VPT improvement. As this was the first pilot study on the therapeutic use of vibrating insoles in a group of patients, the optimal dosage of the intervention was not determined. This could also be of value in future research.

## 5. Conclusions

This is the first study demonstrating that the beneficial effects on the vibration perception threshold of vibratory stimulation can be retained for at least a period of one month after the cessation of the intervention. No other non-pharmacological treatment for the loss of pedal sensation in patients with Type II DM has shown similar longer-term results. The retention of the effect over time cannot currently be explained, but future research on the reasons for the retention and potential of vibratory stimulation for ulcer prevention is warranted.

## Figures and Tables

**Figure 1 sensors-24-03131-f001:**
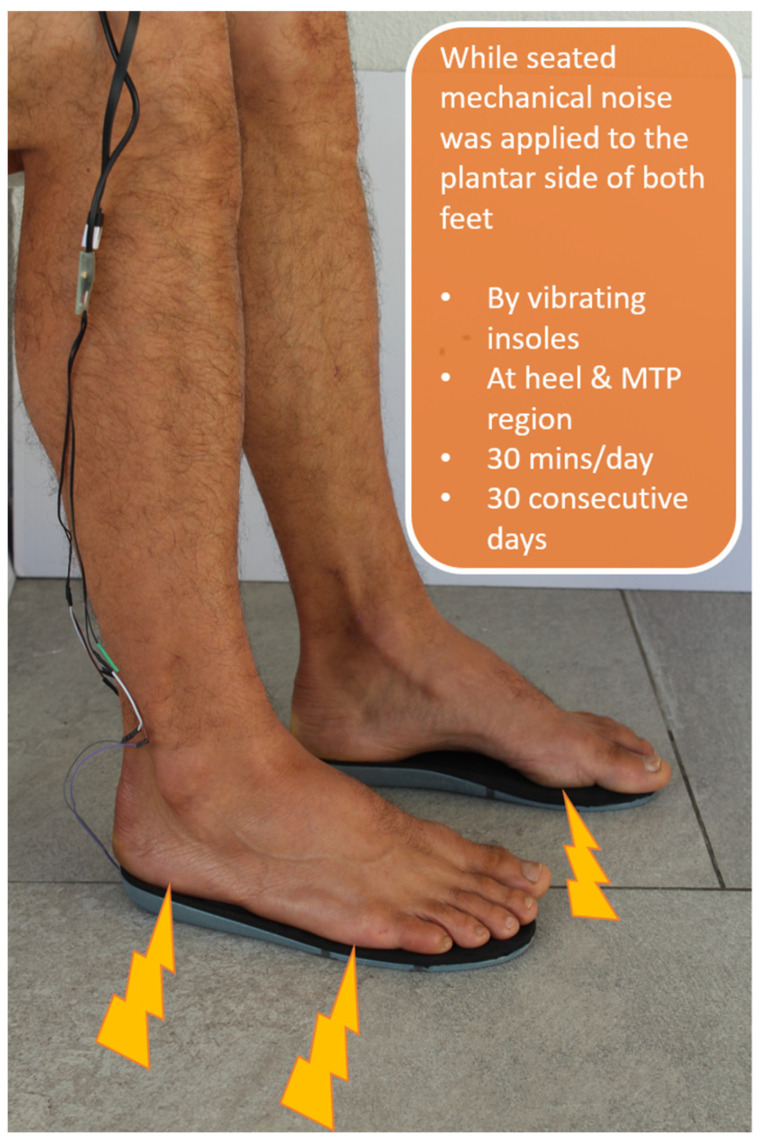
Overview of the intervention.

**Figure 2 sensors-24-03131-f002:**
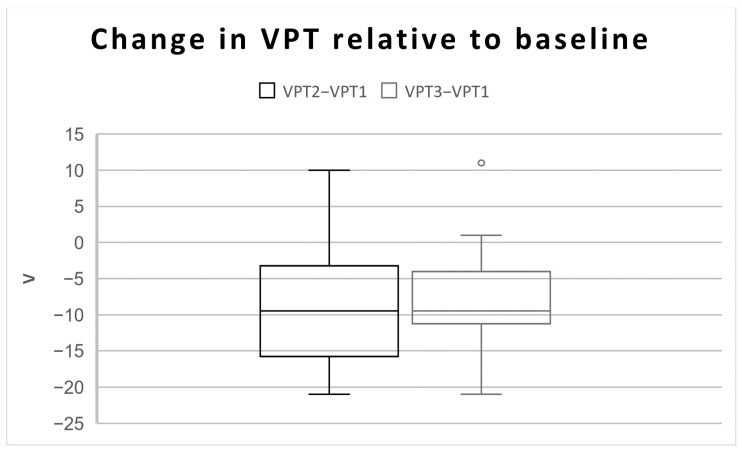
Box plot of the change in VPT between pre-intervention and both post-intervention and post-washout.

**Table 1 sensors-24-03131-t001:** Vibration perception thresholds and pairwise comparisons.

	VPT (Volts)	Pairs	Mean Difference between Measurements (Volt)	Standard Error	*p*-Value	Effect Size (Cohen’s d)
VPT 1 Baseline	34 (SD ± 8.2)	VPT 1–VPT 2	8.5	1.81	* 0.001	1.0
VPT2 Post-intervention	25.5 (SD ± 7.4)	VPT 2–VPT 3	0.3	1.72	1.00	0.1
VPT 3 Post-washout	25.8 (SD ± 8.2)	VPT 1–VPT 3	8.2	1.8	* <0.001	1.1

* Statistically significant.

## Data Availability

Data are contained within the article.

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
