# Peer review of "Retention of Improved Plantar Sensation in Patients with Type II Diabetes Mellitus and Sensory Peripheral Neuropathy after One Month of Vibrating Insole Therapy: A Pilot Study"

_sensors, 2024, doi:10.3390/s24103131_

Round 1

Reviewer 1 Report

Comments and Suggestions for Authors

The main purpose of the work was to determine if the repetitive therapeutic application of mechanical noise applied to the plantar surface of persons with Type II DM could improve VPT and if the improvement was maintained over time.

I have studied the research in detail. I thank the authors for their efforts, this research is original and relevant.

Abstract: structure the abstract according to the journal's instructions: the abstract should be a total of about 200 words maximum. The abstract should be a single paragraph and should follow the style of structured abstracts, but without headings: 1) Background: Place the question addressed in a broad context and highlight the purpose of the study; 2) Methods: Describe briefly the main methods or treatments applied. Include any relevant preregistration numbers, and species and strains of any animals used; 3) Results: Summarize the article's main findings; and 4) Conclusion: Indicate the main conclusions or interpretations.

The introduction section is fairly weak, there is no reference to state-of-the-art of mechanical stimulation/noise applied to the plantar surface using insole. Even if there are no studies about non-pharmacological interventions that have shown long-term efficacy at reducing VPT in people with diabetes, i suggest in order to strengthen the rationale, to cite other studies that have stimulated the plantar surface of the foot through insoles, obtaining positive clinical results (such as DOI: 10.3390/brainsci12121669 and others)

Results:

insert a box plot to make the results easier to read

Discussion:

Authors should add a limitations section. The small number of the sample analyzed (22 participants)  does not support the conclusions

The paper should be considered as a pilot study and accordingly change the title

Comments on the Quality of English Language

minor editing

Author Response

Please see the attachment. I have attached a point by point response to reviewer 1's comments, as opposed to a combined response. 

Reviewer 2 Report

Comments and Suggestions for Authors

The paper presents interesting findings regarding the therapeutic efficacy of vibrating insoles in improving plantar sensation among individuals afflicted with diabetic sensory peripheral neuropathy. By employing a one-month intervention protocol, the authors assessed the impact of vibrating insoles on the vibration perception threshold (VPT), a good indicator of sensory function. This topic holds significant clinical relevance, particularly within the domain of diabetic foot care. While the results outlined in the paper are promising, there are notable weaknesses that warrant attention and resolution:

(1) The abstract needs a thorough rewrite to enhance clarity and ensure a well-organized presentation of the background, methodology, results, and conclusion.

(2) Has there been any prior research exploring improvements in vibration perception threshold resulting from the utilization of vibrating insoles?

(3) How would you compare your findings with existing related works, for example, the work by Cham et al. ("The Effects of Vibro-medical Insole on Vibrotactile Sensation in Diabetic Patients with Mild-to-Moderate Peripheral Neuropathy " DOI: 10.1007/s10072-018-3318-1)?

(4) Could you please provide a comparison between the improvement in Vibration Perception Threshold (VPT) observed in your study, which employs vibrating insoles, and the findings from an existing study focusing on mechanical noise? (Please note that the comparison should consider only short-term results, assuming long-term mechanical noise studies are not available.)

(5) The authors claimed that the improvement in Vibration Perception Threshold (VPT) is long-lasting, making a bold claim despite the available evidence being limited to observations made after just one month.

(6) Authors are advised to pay attention to typographical errors (i.e., repeating words in line 20, double-check whether the meaning of the sentence in lines 108-109 contradicts the subsequent sentence), spelling inconsistencies, and grammatical mistakes.

Comments on the Quality of English Language

Authors are advised to pay attention to typographical errors, spelling inconsistencies, and grammatical mistakes.

Author Response

I have now uploaded a point-by point response to reviewer 2's comments. 

Round 2

Reviewer 1 Report

Comments and Suggestions for Authors

Acceptable for publication if appropriate for the editor and other reviewers.

Comments on the Quality of English Language

minor